# Mesenchymal-Stem-Cell-Derived Extracellular Vesicles Attenuate Brain Injury in *Escherichia coli* Meningitis in Newborn Rats

**DOI:** 10.3390/life12071030

**Published:** 2022-07-11

**Authors:** Young-Eun Kim, So-Yoon Ahn, Won-Soon Park, Dong-Kyung Sung, Se-In Sung, Mi-Sun Yang, Yun-Sil Chang

**Affiliations:** 1Cell and Gene Therapy Institute, Samsung Medical Center, Seoul 06351, Korea; duddms920@skku.edu (Y.-E.K.); wonspark@skku.edu (W.-S.P.); 2Department of Health Sciences and Technology, Samsung Advanced Institute for Health Sciences & Technology (SAIHST), Samsung Medical Center, Seoul 06351, Korea; 3Department of Pediatrics, Samsung Medical Center, Sungkyunkwan University School of Medicine, Seoul 06351, Korea; soyoon.ahn@samsung.com (S.-Y.A.); dbible@skku.edu (D.-K.S.); sein.sung@samsung.com (S.-I.S.); misun.yang@samsung.com (M.-S.Y.)

**Keywords:** mesenchymal stem cell transplantation, exosomes, extracellular vesicles, meningitis, bacterial infections, newborns

## Abstract

We recently reported that transplantation of mesenchymal stem cells (MSCs) significantly reduced bacterial growth and brain injury in neonatal meningitis induced by *Escherichia coli* (*E. coli*) infection in newborn rats. As a next step, to verify whether the MSCs protect against brain injury in a paracrine manner, this study was designed to estimate the efficacy of MSC-derived extracellular vesicles (MSC-EVs) in *E. coli* meningitis in newborn rats. *E. coli* meningitis was induced without concomitant bacteremia by the intra-cerebroventricular injection of 5 × 10^2^ colony-forming units of K1 (-) *E. coli* in rats, at postnatal day 11. MSC-EVs were intra-cerebroventricularly transplanted 6 h after the induction of meningitis, and antibiotics were administered for three consecutive days starting at 24 h after the induction of meningitis. The increase in bacterial growth in the cerebrospinal fluid measured at 24 h after the meningitis induction was not significantly reduced following MSC-EV transplantation. However, an increase in brain cell death, reactive gliosis, and inflammation following meningitis were significantly attenuated after MSC-EV transplantation. Taken together, our results indicate that MSCs show anti-apoptotic, anti-gliosis, and anti-inflammatory, but not antibacterial effects, in an EV-mediated paracrine manner in *E. coli*-induced neonatal meningitis.

## 1. Introduction

Bacterial meningitis is a devastating brain infection that occurs more commonly during the neonatal period than in other ages. In neonatal meningitis, Gram-negative bacteria, such as *Escherichia coli* (*E.coli*), are the most commonly identified organisms, followed by group B streptococci [1]. *E. coli* meningitis can be treated with antibiotics. However, antibiotic effectiveness is limited when antibiotic resistance occurs, and long-term sequelae remain in survivors, particularly in newborns and premature infants. Despite rapid diagnosis and active antibiotic treatment, bacterial meningitis still contributes to high mortality and severe complications, including focal neurological deficits, hydrocephalus, and mental retardation [2]. Therefore, new and effective treatment methods are urgently needed.

Bacterial meningitis leads to inflammation, elevating pro-inflammatory cytokines within the cerebrospinal fluid (CSF) and adjacent brain tissue. These inflammatory processes contribute to the progression of brain injury, including reactive gliosis and cell death in the developing newborn brain [3,4]. Recently, in preclinical and clinical studies, MSC therapy has been applied to various inflammatory diseases, such as *E. coli*-induced acute lung injury [5], intraventricular hemorrhage [6], and hypoxic ischemic encephalopathy [7], because of the MSCs’ anti-inflammatory and cell protective properties mediated by secretion of various immunomodulatory factors and growth factors. In our previous report, the antibacterial effect of MSCs was also demonstrated in *E. coli*-induced neonatal meningitis and pneumonia rodent models [5,8]. These findings suggest that MSC therapy may be a promising and novel method for treating infectious diseases. Transplanted MSCs in host tissue show efficacy in a paracrine manner by secreting EVs containing immunomodulatory factors and growth factors, such as indoleamine 2,3-dioxygenase (IDO), vascular endothelial growth factor (VEGF), and miRNAs [9,10]. However, the efficacy of MSC-derived EVs has not been evaluated in an in vivo neonatal meningitis model. Thus, in this study, we aimed to investigate whether MSC-derived extracellular vesicles (EVs) have therapeutic efficacy via antibacterial, anti-inflammatory, and cell-protective properties in an *E. coli*-induced neonatal meningitis rat model [11].

## 2. Materials and Methods

### 2.1. MSC-Derived EV Preparation

The study was approved by the Institutional Review Board of Samsung Medical Center. Human Wharton’s jelly-derived MSCs were provided by the good manufacturing facility at the Samsung Cell and Gene Therapy Institute. EVs were collected from the culture media of human Wharton’s jelly-derived MSCs after thrombin preconditioning, a method reported to boost the MSC production of EV and their cargo contents [12,13]. Briefly, MSCs were cultured in MEMα with 10% fetal bovine serum (FBS) in a humidified incubator in 5% CO_2_ at 37 °C. At passage 6, the cells were washed in phosphate buffered saline three times to remove contaminating FBS-derived exosomes. Subsequently, during the thrombin preconditioning, the cells were incubated with fresh serum-free MEMα supplemented with recombinant thrombin, as detailed in our previous report [12]. EVs were isolated from the conditioned medium of MSCs by centrifugation at 3000 rpm for 30 min at 4 °C (Eppendorf, Hamburg, Germany) to remove cellular debris, and then at 100,000× *g* for 120 min at 4 °C (Beckman, Brea, CA, USA) to sediment the EVs. As described in our previous report [10], the purity and characterization of EVs were checked by their images using scanning electron microscopy (Zeiss Auriga Workstation, Oberkochen, Germnay) and transmission electron microscopy (FEI, Hillsboro, OR, USA), the particle size distribution and number of EVs was measured using a NanoSightN300 and Nanoparticle Tracking Analysis (NTA) software, and the exosomal protein markers such as CD9, CD64, and CD81 using a Western blot (Appendix A).

### 2.2. Bacterial Preparation

An un-encapsulated mutant strain of *E. coli* possessing the K1 capsular polysaccharide C5 (EC5ME; serotype 018:K1:H7), was cultured as the source of induction of neonatal meningitis, as previously described [8]. The *E. coli* strain was a gift from Professor Kwang Sik Kim, Johns Hopkins University, MD, USA. *E. coli* in the mid-log phase was prepared at a concentration of 5 × 10^2^ colony-forming units (CFU) in 10 μL of normal saline. The concentration of inoculated *E. coli* was confirmed by serial dilutions of bacterial cultures on solid agar media.

### 2.3. Animal Model of Neonatal E. coli Meningitis

The experimental protocols were reviewed and approved by the Institutional Animal Care and Use Committee (IACUC) of the Samsung Biomedical Research Institute. This study followed the institutional and National Institutes of Health guidelines for laboratory animal care. All animal procedures were performed in an AAALAC-accredited specific-pathogen-free facility at the Samsung Biomedical Research Institute. As previously described [8], newborn Sparague–Dawley rats (Orient Co., Seoul, Korea) at postnatal day 11 were anesthetized with 1.5–3.0% isoflurane in oxygen-enriched air and inoculated with 5 × 10^2^ CFU of *E. coli* (EC5ME) in 10 μL of normal saline into the left cerebroventricle using a stereotaxic frame (MyNeurolab, St. Louis, MO, USA; coordinates: x = −0.5 mm, y = +1.3 mm, z = +3.0 mm relative to the bregma). An equal volume of normal saline was administered to the control group in the same manner. For transplantation of MSC-EVs, the dose of MSC-EVs was determined with reference to the results of our previous studies [8]. MSC-EVs, isolated from 1 × 10^5^ MSCs, in 10 μL of normal saline were transplanted into the right cerebroventricle using a stereotaxic frame (x = +0.5 mm, y  =  +1.3 mm, z = +3.0 mm relative to the bregma), at 6 h after induction of meningitis. Equal volumes of saline were administered to the non-transplanted control group in the same manner. *E. coli* or MSC-EVs in 10 μL of normal saline was injected at a very low rate (10 μL/min) into the cerebroventricle under stereotaxic guidance. No mortality was associated with any of the procedures. Antibiotics (ampicillin; Samyang Anipharm Co., Ltd., Seoul, Korea) were intraperitoneally administered (200 mg/kg/day) for three consecutive days, starting 24 h after the induction of meningitis. Rat pups were randomly assigned to each group (NC, normal control = 6; MC, meningitis control = 12; and meningitis with transplantation of MSC-EVs, ME = 11). CSF was collected from the cisterna magna and brain tissue was extracted at P17 under deep pentobarbital anesthesia (60 mg/kg, intraperitoneal) as previously described [8]. Brain tissue was fixed with 4% PFA for 24 h for histological analysis and CSF was snap-frozen and stored at −70 °C for biochemical analysis. The experimental schedules are shown in detail in Appendix A.

### 2.4. Bacterial Quantification

CSF and blood were obtained to measure the bacterial titer at 6 h, 24 h, and 6 days after bacterial inoculation for meningitis induction. Quantification of CFU from *E. coli* in the CSF was performed at dilutions of 10^−4^–10^−8^, plated on solid agar media. The bacterial CFU in the blood were measured without dilution, on solid agar and broth media.

### 2.5. Terminal Deoxynucleotidyl Transferase dUTP Nick End Labeling (TUNEL) Assay

To detect dead cells in brain tissues, the TUNEL assay (kit G3250, Promega, Madison, WI, USA) was performed on deparaffinized 5 μm-thick brain sections, according to the manufacturer’s protocol. The number of TUNEL-positive cells were determined in the periventricular area (+1.8 mm to +0.95/bregma); two fields (right and left ventricle) were captured in one brain slice, and three brain slices were analyzed by a blinded evaluator.

### 2.6. Immunohistochemistry

To detect reactive gliosis and microglia, immunohistochemical analysis of glial fibrillary acidic protein (GFAP; Dako, Glostrup, Denmark) and ED1 (Abcam, Cambridge, UK) was performed on deparaffinized 5 μm-thick brain sections. The light intensity of the GFAP-positive area and the number of ED1-positive cells were measured in the periventricular area (+1.8 mm to +0.95/bregma); two fields (right and left ventricle) were captured in one brain slice, and three brain slices were analyzed by a blinded evaluator.

### 2.7. Enzyme-Linked Immunosorbent Assay

Levels of inflammatory cytokines such as interleukin (IL)-1α, IL-1β, IL-6, and tumor necrosis factor (TNF)-α, were measured in CSF using a MILLIPLEX MAP ELISA Kit according to the manufacturer’s protocol (EMD Millipore, Billerica, MA, USA).

### 2.8. Statistical Analyses

Data are expressed as mean ± standard error of the mean. All data had significant normal distribution (Shapiro–Wilk normality test, *p* > 0.05). For continuous variables, statistical comparisons between groups were performed using one-way analysis of variance and Tukey’s post hoc analysis. A *p*-value of < 0.05 was considered statistically significant.

## 3. Results

### 3.1. Bacterial Count

After meningitis induction, bacterial CFU in CSF were measured in the meningitis control and in the meningitis with transplantation of MSC-derived EVs (MSC-EVs) groups, before antibiotic administration (6 h and 24 h) and after antibiotic administration (6 day) (Figure 1). Bacterial CFU were not detected in normal control group at 6 h, 24 h, and 6 days after the sham procedure. Before transplantation of MSC-EVs at 6 h, bacterial CFU were not significantly different between the two groups. However, the bacterial CFU did not change significantly after transplantation of MSC-EVs at 24 h. Bacterial CFU were not detected on day 6 after inducing meningitis because this was after antibiotic administration. In blood cultures, bacterial CFU were not detected at 6 h, 24 h, and 6 days after meningitis infection. In an in vitro study, we confirmed that MSCs, but not MSC-EVs, were effective for bacterial clearance in the culture media (Appendix A).

### 3.2. Cell Death and Reactive Gliosis

The number of dead cells, marked as TUNEL-positive cells, significantly increased after meningitis induction, compared with that in normal controls (Figure 2). However, the increase in cell death was significantly reduced after transplantation of MSC-EVs.

The levels of reactive gliosis, marked by the GFAP-positive area, were significantly elevated after meningitis induction compared with those in normal controls (Figure 3). However, the increase in reactive gliosis was significantly reduced after transplantation of MSC-EVs.

### 3.3. Brain Inflammation

The number of active microglia, marked as ED1-positive cells, was significantly increased after meningitis induction compared with that in normal controls (Figure 4). However, the increase in the number of active microglia was significantly reduced after the transplantation of MSC-EVs.

The levels of inflammatory cytokines, including interleukin (IL)-1α, IL-1β, IL-6, and TNF-α, in brain tissues were significantly elevated after meningitis induction compared with those in normal controls (Figure 5). The levels of IL-1α and IL-6 were slightly reduced and the levels of IL-1β and TNF-α were significantly reduced after transplantation of MSC-EVs.

## 4. Discussion

Previously, we demonstrated the effect of MSCs in inhibiting bacterial growth and attenuating cell death, reactive gliosis, and inflammation, in an *E. coli*-induced neonatal meningitis model [8]. Numerous studies have shown that paracrine mechanisms are key for the therapeutic benefits of MSC-based therapy, based on the fact MSCs secrete various antibacterial and immunomodulatory factors [14]. Thus, in this study, we aimed to investigate the effect of EVs isolated from MSCs on *E. coli*-induced neonatal meningitis in an in vivo model. To develop an ideal animal model that mimics the pathophysiology and histopathological characteristics of neonatal bacterial meningitis, we previously tested the intra-cerebroventricular inoculation of *E. coli* (EC5ME) at a dose of 5 × 10^2^ CFU in postnatal day 11 (P11) newborn rats; this dose significantly reduced survival rate and body and brain weight, and induced brain infarction and inflammation and ensuing brain injury, without concomitant bacteremia [11]. In this study, the previously developed meningitis model was used to evaluate the efficacy of MSC-derived EVs (MSC-EVs). We observed that intra-cerebroventricular transplantation of MSC-EVs significantly attenuated brain inflammation and subsequent brain injuries associated with the meningitis-induced brain such as cell death and reactive gliosis, although it did not significantly inhibit the bacterial growth. This is the first study demonstrating that MSC-EVs are effective in preventing inflammation and protecting brain cells in newborn rats with bacterial meningitis.

Although modern antibiotics have been optimally designed to prevent bacterial meningitis and kill bacteria, their use has limitations, such as development of antibiotic resistance or long-lasting complications in survivors. Bacterial meningitis is caused by bacterial entry into the spinal cord and brain. Usually, initiation of an acute infectious/inflammatory response begins within 3 h, and peaks within 24 h [15,16]. However, the high mortality and resulting brain injury, such as neurological damage, are largely caused by the severe inflammatory response of the host against the bacteria [17,18]. In the developing brain of infants, physical impairment can persist throughout their lives [18]. We started administering antibiotics after 24 h when the inflammatory reaction was already high. Despite eliminating bacteria by administrating antibiotics after 24 h, inflammatory reactions can persist and continuously cause cell death and reactive gliosis after 6 days of inducing bacterial meningitis. Therefore, preventing brain inflammation is an important goal in treating bacterial meningitis in infants. Furthermore, the development of a new adjunctive therapy to attenuate inflammation is urgently needed. In this study, we observed that the use of antibiotics alone was effective in killing bacteria, but not in attenuating inflammation. However, brain inflammation and ensuing brain injuries were significantly attenuated when antibiotics and MSC-EVs were administered together, compared with those with monotherapy with antibiotics. In this regard, MSC-based therapy could be regarded as a novel treatment approach for neonatal meningitis, owing to the anti-inflammatory and immunomodulatory properties of MSCs. To modulate the severity of inflammation, MSCs secrete anti-inflammatory cytokines, such as IL-10, and growth factors, such as BDNF, VEGF, FGF, and TGF-β [19]. Growing evidence indicates that EVs, including exosomes derived from the MSC secretome, contain anti-inflammatory cytokines and growth factors [20,21], suggesting that MSC-EVs might be a key element in the anti-inflammatory and immunomodulatory properties of MSCs. In this study, we found that MSC-EVs significantly reduced the number of active microglia, which release inflammatory cytokines promoting the inflammatory response, and the levels of inflammatory cytokines such as IL-1β and TNF-α. Additionally, MSC-EVs significantly attenuated the ensuing brain injuries such as cell death and reactive gliosis, a reaction to brain injury. This suggests that MSC-EVs might be a promising agent for MSC-based cell-free therapy in inflammatory brain injuries. In clinical applications, MSC-EVs have the advantages of being easier and simpler to store and manipulate compared with MSC, and do not have oncogenic properties. That said, MSC-EVs are still as heterogeneous as MSCs, and the production technology for a high yield and purity of MSC-EVs for clinical applications has not yet been standardized.

According to previous studies on MSCs’ antibacterial effect, MSCs showed a significant effect on bacterial clearance in an *E. coli* infection model, via secreting anti-microbial peptides, such as β-defensin2, LL-37, and lipocalin2 [5,22,23]. However, a limitation of our study is that we did not observe a significant effect of MSC-EVs on bacterial clearance in *E. coli*-induced bacterial meningitis, even though we had observed the bacterial clearance effect of MSCs in our previous study in an animal model of *E. coli*-induced acute respiratory distress syndrome and neonatal meningitis [5,8]. After infection with *E. coli,* MSCs upregulated expression levels of defense-related genes, including β-defensin2, via TLR2 and TLR4 signaling, and the growth of *E. coli* was significantly inhibited by co-culture with MSCs or by treatment of conditioned media of bacteria-preconditioned MSCs [5]. However, in this study, we used thrombin-preconditioned MSCs, rather than bacteria- or LPS-preconditioned MSCs to boost the MSC production of EVs and to enhance the anti-inflammatory effect [13,24,25]. In other words, MSCs optimize their beneficial paracrine action depending on culture milieu or host milieu. Thus, the MSC-EVs in this study may not have a significant antibacterial effect, although they were effective in anti-inflammation. Whether MSC-EVs isolated from preconditioned MSCs with infective stimuli have antibacterial effects through defense-related paracrine mechanisms should be validated in the bacterial meningitis model in further study. Additionally, whether MSCs’ antibacterial peptides and molecules are secreted via EVs or directly secreted has not been investigated, so further study will be necessary to clarify this. The important finding in the study was that MSC-EVs maintain MSCs’ capacity to attenuate cell death, reactive gliosis, and inflammation in bacterial meningitis in newborn rats. In this study, our data suggest that transplantation of MSC-EVs may be a candidate for combination treatment with antibiotics in bacterial meningitis in neonates.

## 5. Conclusions

Intra-cerebroventricular transplantation of MSC-EVs is effective in protecting newborn rats against bacterial-meningitis-induced brain injury without concomitant bacteremia. MSC-EVs did not significantly inhibit bacterial growth, but significantly attenuated brain cell death, reactive gliosis, and inflammation in newborn rats with neonatal meningitis.

## 6. Patents

Y.-S.C., W.-S.P., D.-K.S. and S.-Y.A. declare potential conflicts of interest arising from a filed or issued patent titled “Method for promoting generation of stem cell-derived exosome by using thrombin. (10-1661448, PCT/KR2014/012291)” and “Composition for treating inflammatory brain disease comprising stem-cell-derived exosome as an active ingredient. (10-1661847, PCT/KR2015/002640)”, as co-inventors, not as patentees.

## Figures and Tables

**Figure 1 life-12-01030-f001:**
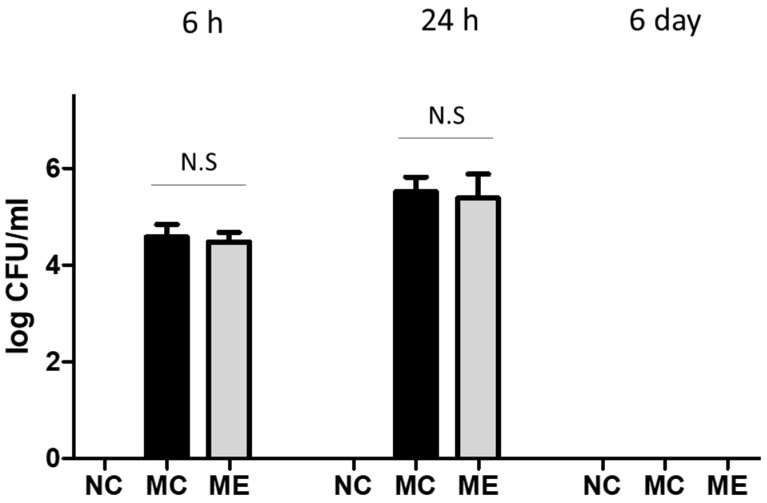
Transplantation of MSC-EVs did not significantly affect bacterial clearance in cerebrospinal fluid (CSF). Colony-forming units (CFU) of cultured *E. coli* in the CSF at 6 h, 24 h, and 6 days after meningitis induction (*n* = 6, 12, and 11 in the NC group, MC group, and ME group, respectively). Data are expressed as mean ± SEM. Significant probability with critical level set at 0.05; N.S, not significant (*p* > 0.05). NC, normal control group; MC, meningitis control group; ME, meningitis with transplantation of MSC-EVs group.

**Figure 2 life-12-01030-f002:**
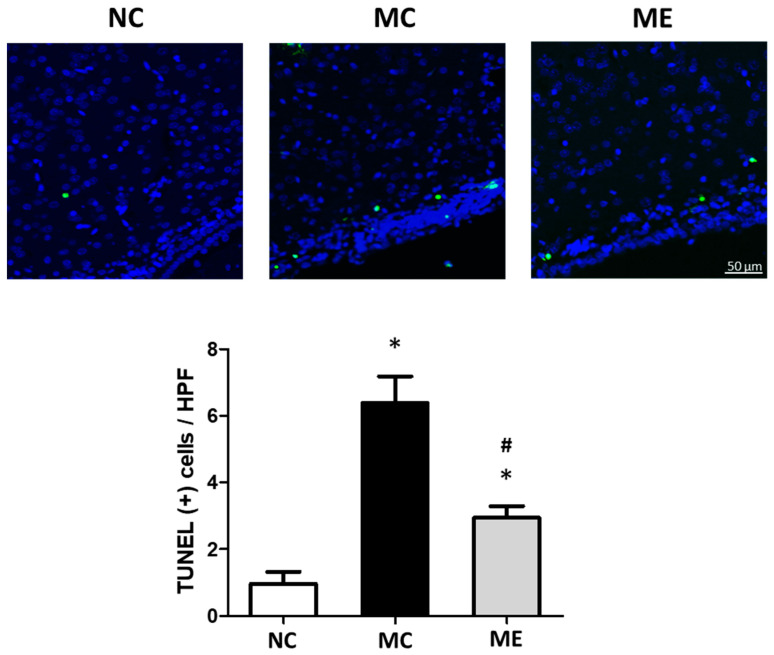
Transplantation of MSC-EVs attenuated brain cell death 6 days after meningitis induction. Representative fluorescence microscope images of the periventricular area with staining for terminal deoxynucleotidyl transferase dUTP nick end (TUNEL; green) and DAPI (blue) (upper panel), and average number of TUNEL-positive cells in high-power field (HPF) (lower panel). Only TUNEL and DAPI double-positive nuclei were counted (*n* = 6, 12, and 11 in the NC group, MC group, and ME group, respectively). Original magnification, ×200; scale bars, 50 μm. Data are expressed as mean ± SEM. NC, normal control group; MC, meningitis control group; ME, meningitis with transplantation of MSC-EVs group. * *p* < 0.05 vs. NC, # *p* < 0.05 vs. EC.

**Figure 3 life-12-01030-f003:**
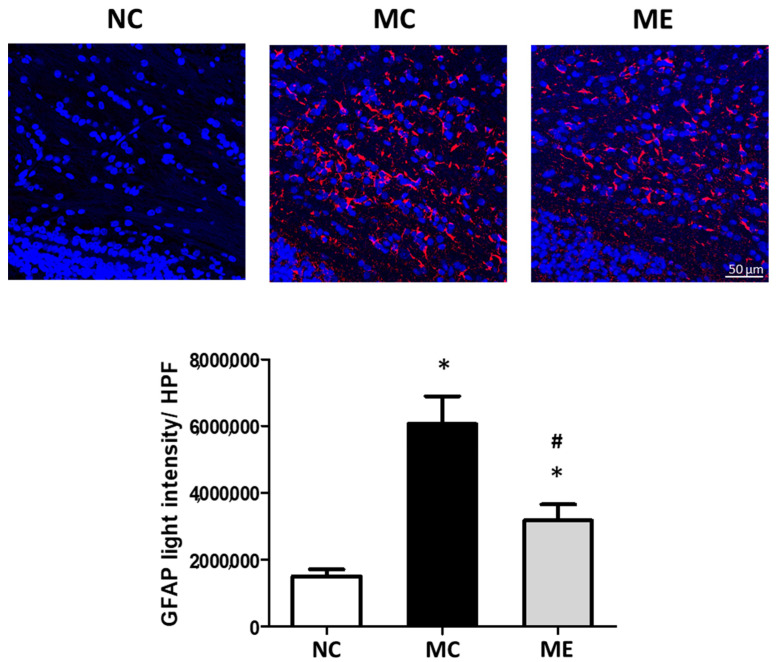
Transplantation of MSC-EVs attenuated reactive gliosis 6 days after meningitis induction. Representative fluorescence microscope images of the periventricular area with staining for glial fibrillary acidic protein (GFAP; red) and DAPI (blue) (upper panel), and average light intensity of the GFAP-positive area in high-power field (HPF) (lower panel) (*n* = 6, 12, and 11 in the NC group, MC group, and ME group, respectively). Original magnification, ×200; scale bars, 50 μm. Data are expressed as mean ± SEM. NC, normal control group; MC, meningitis control group; ME, meningitis with transplantation of MSC-EVs group. * *p* < 0.05 vs. NC, # *p* < 0.05 vs. EC.

**Figure 4 life-12-01030-f004:**
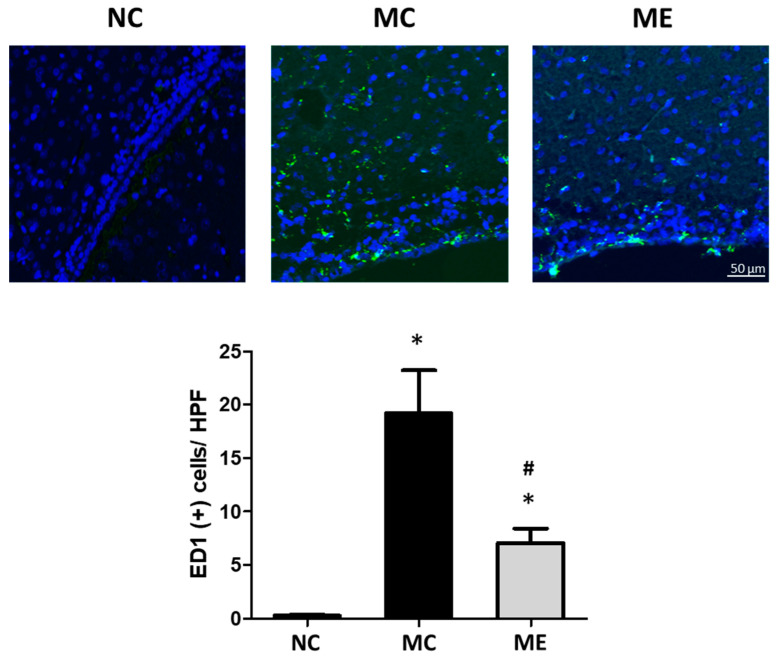
Transplantation of MSC-EVs attenuated the number of active microglia 6 days after meningitis induction. Representative fluorescence microscope images of the periventricular area with staining for ED1 (green) and DAPI (blue) (upper panel), and average number of ED1-positive cells in high-power field (HPF) (lower panel). Only ED1-positive cells surrounding a DAPI-stained nucleus were counted (*n* = 6, 12, and 11 in the NC group, MC group, and ME group, respectively). Original magnification, ×200; scale bars, 50 μm. Data are expressed as mean ± SEM. NC, normal control group; MC, meningitis control group; ME, meningitis with transplantation of MSC-EVs group. * *p* < 0.05 vs. NC, # *p* < 0.05 vs. EC.

**Figure 5 life-12-01030-f005:**
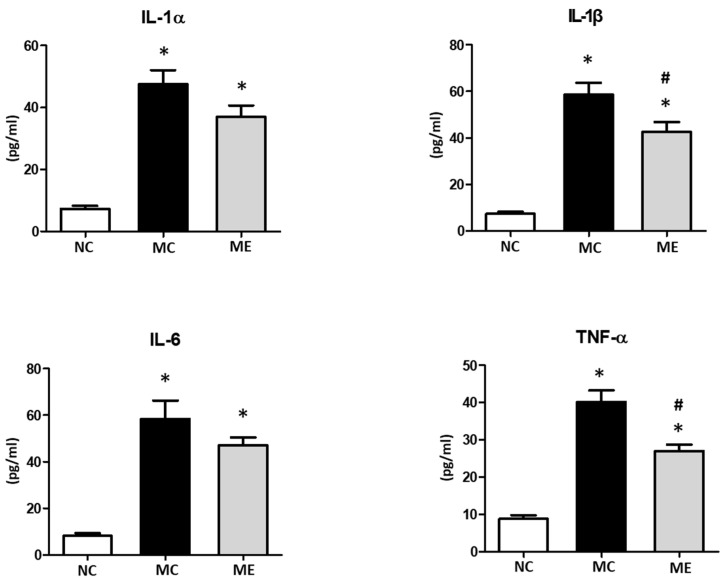
Transplantation of MSC-EVs attenuated brain inflammatory cytokines 6 days after meningitis induction. Levels of interleukin (IL)-1α, IL-1β, IL-6, and tumor necrosis factor (TNF)-α in CSF (*n* = 6, 12, and 11 in the NC group, MC group, and ME group, respectively). Data are expressed as mean ± SEM. NC, normal control group; MC, meningitis control group; ME, meningitis with transplantation of MSC-EVs group. * *p* < 0.05 vs. NC, # *p* < 0.05 vs. EC.

## Data Availability

Not applicable.

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
