# Peer review of "Mesenchymal-Stem-Cell-Derived Extracellular Vesicles Attenuate Brain Injury in Escherichia coli Meningitis in Newborn Rats"

_life, 2022, doi:10.3390/life12071030_

Round 1
Reviewer 1 Report
Dear authors:
I have a big doubt, in an 11-day-old animal, is it possible to inject 10 microliters intracerebral ventricular? And then more treatment?
How long did it take to inject this volume?
How were the animals anesthetized?
How many animals did you use?
How was the postoperative period?
I ask about the isolation of the vesicles:
I believe 100,000 RPM is a lot. Wouldn't it be 100,000xg?
I suggest that you have briefly described the isolation, and characterization of the vesicles, some imaging or cytometry.
About the cells: What medium was used? With fetal bovine serum, how many percent, how did you make the vesicles in the serum not a contaminating factor? What passage of cells did you use?
About the microscopy images:
The quality of the images is not good. And the size should also be bigger.
I suggest presenting a schematic of the location where the images were taken.
I suggest adding the cell font in the title and throughout the text.
I suggest a review of these points.
Yours sincerely!
Author Response
- Reviewer #1
1-1) The reviewer stated, “I have a big doubt, in an 11-day-old animal, is it possible to inject 10 microliters intracerebral ventricular? And then more treatment?”
-> We have previously conducted multiple experiments and have published studies regarding the efficacy of MSCs transplanted into the cerebroventricle in a volume of 10 μl (normal saline) in postnatal day (P) 6, P8, and P11 rat pups (Cell Transplant. 2016;25(6):1131-44, Sci Rep. 2018 May 16;8(1):7665, Pediatr Res. 2018 Nov;84(5):778-785). With this experience of performing similar experiments, in the present study, we were able to inject E. coli in 10 μl of normal saline into the left cerebroventricle, and then 6 hours later, MSC-EVs in 10 of μl normal saline into the right cerebroventricle in the P11 rat pups successfully.
1-2) The reviewer stated, “How long did it take to inject this volume?”
-> E. coli (5 × 102 CFU) or MSC-EVs (isolated from 1 × 105 MSCs) in 10 μl of normal saline was injected at a very low rate (10 μl/min) into the cerebroventricle under stereotaxic guidance. We have incorporated this into the Methods section 2.3 (Lines 107–108).
1-3) The reviewer stated, “How were the animals anesthetized?”
-> During experimental animal modeling, the rat pups were anesthetized with 1.5–3.0% isoflurane in oxygen-enriched air. We have incorporated this into the Methods section 2.3. (Line 97)
1-4) The reviewer stated, “How many animals did you use?”
-> We used 29 rat pups (normal control = 6, meningitis control = 12, and meningitis with transplantation of MSC-EVs = 11, respectively), as described in the Methods section 2.3 (Lines 112–113). To specify the number of animals we used, further details have been added in the figure legends of Figures 1–5.
1-5) The reviewer stated, “How was the postoperative period?”
-> After inducing meningitis, the rat pups were returned to the cage with the mother rat. Subsequently, transplantation of MSCs was followed 6 h later. We have described the experimental plan in detail in Supplementary Figure S2.
1-6) The reviewer stated, “I ask about the isolation of the vesicles: I believe 100,000 RPM is a lot. Wouldn't it be 100,000xg?”
-> We thank the reviewer for the valuable comments on our manuscript. We have corrected “1000,000 rpm” to “100,000 × g” in the Methods section of the revised manuscript (Line 75).
1-7) The reviewer stated, “I suggest that you have briefly described the isolation, and characterization of the vesicles, some imaging or cytometry.”
-> Briefly, EVs were isolated from the conditioned medium of MSCs by centrifugation at 3,000 rpm for 30 min at 4°C to remove cellular debris, and then at 100,000 × g for 120 min at 4°C to sediment the EVs. In our revised manuscript, the electron microscopy images, the particle size distribution of EVs, and western blot images of the exosomal protein markers such as CD9, CD64, and CD81 have been added in Supplementary Figure S1. We have incorporated these methodological details into the Methods section 2.3 in the revised version of our manuscript (Lines 72–81).
1-8) The reviewer stated, “About the cells: What medium was used? With fetal bovine serum, how many percent, how did you make the vesicles in the serum not a contaminating factor? What passage of cells did you use?”
-> MSCs were cultured in MEMα with 10% fetal bovine serum (FBS) in a humidified incubator in 5% CO2 at 37 °C. At passage 6, the cells were washed in phosphate- buffered saline three times to remove contaminating FBS-derived exosomes. Subsequently, during the thrombin-preconditioning, the cells were incubated with fresh serum-free MEMα supplemented with recombinant thrombin, as detailed in our previous report (New Reference #12). We have incorporated these details into the Methods section 2.1. (Lines 68–72)
1-9) The reviewer stated, “About the microscopy images: the quality of the images is not good. And the size should also be bigger.”
-> As recommended, the images have been improved in terms of quality and have been enlarged, in our revised manuscript.
1-10) The reviewer stated, “I suggest presenting a schematic of the location where the images were taken.”
-> The fluorescence microscopy images were captured at the periventricular zone; two fields (right and left ventricle) were captured in one brain slice, and three brain slices were analyzed. We have incorporated these details into the Methods section 2.5. (Lines 126–127) and 2.6. (Lines 134–135)
Additionally, as recommended, a schematic diagram of the locations where the images were captured has been highlighted using red squares in Supplementary Figure S2.
1-11) The reviewer stated, “I suggest adding the cell font in the title and throughout the text.”
-> We sincerely regret that we did not use the cell font the reviewer recommended. We could not find a way to use the cell font in this manuscript. However, according to the author's instructions, we used the font provided by the Microsoft Word template on “Life” website.

Reviewer 2 Report
I have few questions/comments:
· How was the purity of the MCS-Evs checked and were they cleaned additionally?
· where in Fig. 1 (or in the text) is a reference to the control (normal saline)? Was such an control done, and if so, please include that in the results.
· Figure 1 needs to be corrected, it looks like the Authors forgot to add the bars for 6 days.
· whether antibiotics were also given after 6 and 24 hours?
· How can the Authors explain the fact that after 6 days there was no E. coli (Fig. 1), but the effect concerning cell death, brain inflammation, cell death and reactive gliosis was observed?
Please include answers to these questions and comments in the text.
Author Response
- Reviewer #2
2-1) The reviewer stated, “How was the purity of the MCS-EVs checked and were they cleaned additionally?”
-> The purity and characterization of EVs were checked by their size and number of EVs as measured using a NanoSightNS300 and Nanoparticle Tracking Analysis software. The electron microscopic images, the particle size distribution of EVs, and western blots images of the exosomal protein markers such as CD9, CD64, and CD81 have been added in Supplementary Figure S1, in the revised version of our manuscript (Lines 76–81).
2-2) The reviewer stated, “Where in Fig. 1 (or in the text) is a reference to the control (normal saline)? Was such a control done, and if so, please include that in the results.”
-> Bacterial CFUs were not detected in normal control (normal saline) group at 6 h, 24 h, and 6 days after the sham procedure. We have added the details regarding the normal control group in Figure 1, and incorporated this into the Results section 3.1, in our revised manuscript (Lines 152–153).
2-3) The reviewer stated, “Figure 1 needs to be corrected, it looks like the Authors forgot to add the bars for 6 days.”
-> Bacterial CFUs were not detected on day 6 after meningitis because this was after antibiotic administration. The bars on day 6 represent ‘zero (0)’ in all experimental groups, as described in the Result section (Lines 155–157),
2-4) The reviewer stated, “Whether antibiotics were also given after 6 and 24 hours?
-> Antibiotics were not administered after 6 hours. After 24 hours, CSF was tapped for bacterial CFU measurement, and antibiotics were administrated for three consecutive days (ampicillin; 200 mg/kg/day). To simplify the experimental schedule, the protocol of antibiotics administration has been added in Supplementary Figure S2.
2-5) The reviewer stated, “How can the Authors explain the fact that after 6 days there was no E. coli (Fig. 1), but the effect concerning cell death, brain inflammation, cell death and reactive gliosis was observed?”
-> We sincerely thank the reviewer for allowing us the opportunity to explain the meningitis model in this study in detail. Usually, the initiation of acute infectious/inflammatory response begins within 3 h, and peaks within 24 h (Front Pediatr. 2022 Mar 8;10:840288, Cell Transplant. 2016;25(6):1131-44). When newborns develop bacterial meningitis, the resulting brain injuries are largely caused by the severe inflammatory response of the host against the bacteria (Front Microbiol. 2019 Mar 25;10:576). We started administering antibiotics after 24 h when inflammatory reaction was already high. Despite eliminating bacteria by administrating antibiotics after 24 h, inflammatory reactions can persist and continuously cause cell death and reactive gliosis after 6 days of inducing bacterial meningitis. Therefore, preventing brain inflammation is an important goal in treating bacterial meningitis in infants. Furthermore, the development of a new adjunctive therapy to attenuate inflammation is urgently needed. We have incorporated these details into the Discussion section (Lines 238–249).

Round 2
Reviewer 1 Report
Accept in present form!
Author Response
Dear Reviewer#1
We appreciate the reviewer's valuable comment so we can improve our work, and the reviewer's acceptance in the manuscript.
Sincerely,
Yun Sil Chang, MD, PhD
Department of Pediatrics, Samsung Medical Center, Sungkyunkwan University school of Medicine, 50 Irwon-dong, Gangnam-gu, Seoul 135-710, Korea
Tel: +82.2-3410-3528
Reviewer 2 Report
Few editorial errors must be corrected, e.g., no periods at the end of a sentence. More detailed descriptions of Figure 2 and Figure 4 are needed, description of green signals in the pictures.
Author Response
We appreciate the reviewer’s insightful and helpful comments and have incorporated of the feedback in to the revised manuscript.
- Few editorial errors have been corrected (Lines 28, 119, 193 and 306)
- More detailed descriptions of Figure 2 and Figure 4 have been incorporated in their figure legends (Lines 178-179 and 203-204)
